# Liver, vessel, and tumor segmentation from partially labeled CT and multi-label masked learning

Eirik A. Østmo *[1], Keyur Radiya[1,2], Kristoffer K. Wickstrøm[1], Michael C. Kampffmeyer[1,3], Karl Øyvind Mikalsen[1,2], and Robert Jenssen[1,3,4]

[1]UiT The Arctic University of Norway
[2]University Hospital of North Norway
[3]Norwegian Computing Centre
[4]University of Copenhagen

## Abstract

Accurate delineation of liver parenchyma, intrahepatic vessels, and tumors (LVT) may aid earlier tumor detection, consistent response assessment, and surgical planning for patients with liver cancer. Deep learning (DL) may enable such automated delineation, but available CT datasets are inconsistent and partially labeled, making them unsuited for end-to-end training. We investigate a single-head, 3D segmentation framework that learns from partially labeled data by: (i) loss masking per class or voxel to ignore missing annotations, (ii) using multi-hot targets and the anatomical hierarchy inherent to liver, vessels, and tumors, to handle overlapping structures without class competition. In controlled ablations that simulate partial-label training, this multi-label masked strategy reliably outperforms masked multi-class baselines, avoids precision collapse, and improves tumor overlap and lesion detection sensitivity. Scaling training to multiple partially labeled datasets, the model surpasses full-resolution nnU-Net on an external clinical cohort, with higher tumor and vessel segmentation performance. We conduct a retrospective feasibility analysis on clinical data to illustrate the clinical potential of the LVT application. We find that LVT models may facilitate earlier detection of metastasis, longitudinal size tracking aligned with radiologist measurements, 3D tumor–vessel visualization for surgical planning, and stable inter-phase liver volumetry ($\approx 5\%$ deviation). These results show that multi-label masked learning enables robust, clinically relevant LVT segmentation from partially labeled datasets.

## 1 Introduction

Effective management of liver cancer, including patient follow-up and surgical treatment, relies on patient-specific understanding of the liver parenchyma, intrahepatic vasculature, and tumor burden. Accurate delineation of these structures may enable a range of impactful clinical tasks: earlier and more reliable tumor detection during patient follow-up, objective and consistent longitudinal response assessment, preoperative planning with 3D visualization, and automatic liver volumetry to estimate functional reserve before resections [1–7].

Deep learning (DL) based segmentation models have the potential to automatically produce high-quality segmentations of the liver parenchyma, intrahepatic vessels, and hepatic tumors (LVT) [8, 9]. However, publicly available 3D annotation of liver, vessels, and tumors, to train such models, remains scarce and partially labeled. This has constrained clinically geared liver applications to single-task models, limiting generalizability and complicating clinical deployment.

In this paper, we address the scarcity of fully labeled CT liver, vessel, and tumor segmentation data and the fragmentation of labeled datasets. To improve generalizability with limited data, we leverage a recently proposed augmentation strategy for contrast-enhanced CT liver images called Random windowing [10, 11]. Furthermore, to exploit datasets with partial labels, we explore multiple segmentation strategies capable of learning from partial labels and potentially overlapping structures end-to-end. Ultimately, we try to answer the question: How to leverage partially labeled datasets with overlapping structures in LVT segmentation? We identify that multi-label binary segmentation with a masked loss and multi-hot encoded labels, to allow class overlaps, can balance the loss contribution of partial labels and better learn from overlapping classes (Figure 1).

We demonstrate the effectiveness and scalability of our approach with quantitative evaluation against the nnU-Net baseline [12]. To complement the quantitative evaluation and to demonstrate the clinical potential of automatic DL segmentation of LVT structures, we qualitatively evaluate a clinical case study that highlights the potential of such models.

The case study illustrates clinical feasibility and how automatic LVT predictions could facilitate earlier detection of liver tumors, track tumor size over time comparable to radiologist measurements, and deliver 3D visualizations of tumor–vessel relationships. We also show that automated liver volumes

---

*Corresponding Author.

Proceedings of the 7th Northern Lights Deep Learning Conference (NLDL), PMLR 307, 2026.

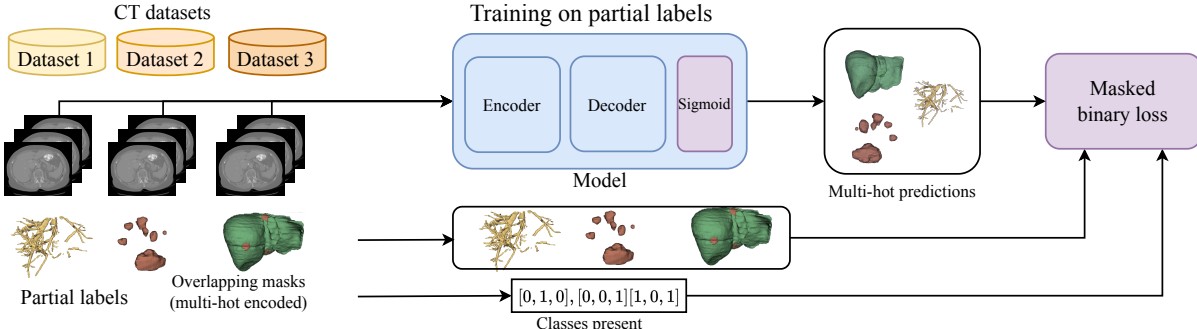

**Figure 1.** A schematic overview displaying our approach for learning robust and simultaneous liver, vessel, and tumor segmentation from multiple partially labeled datasets. Formulating the objective functions for binary segmentation allows us to define overlapping classes and mask the loss based on the missing labels.

remain stable across contrast phases, supporting volumetric assessment in clinical workflows.

Our contributions are twofold:

1. We analyze how to efficiently use partially labeled data with overlapping regions to segment the liver, vessels, and tumors in CT images.

2. We demonstrate clinical feasibility of LVT models through a combination of quantitative evaluation on challenging clinical data and qualitative retrospective cases that highlight earlier tumor detection, longitudinal monitoring, 3D surgical planning, and consistent volumetry.

## 2 Related work

CT-based segmentation of LVT has often been addressed with task-specific models, as many of the impactful and available datasets provide only a subset of these labels. Methodological developments have been driven largely by medical segmentation challenges, and public datasets from these, such as LiTS/MSD Liver for liver and tumor, MSD HepaticVessel for hepatic vessels [8, 9]. Additionally, 3Dircadb provides the complete label set, but on a comparatively small cohort (20 cases) [13].

Architecturally, 3D encoder–decoder CNNs and the self-configuring nnU-Net remain strong baselines for medical 3D applications despite recent advances in vision transformers [12, 14–16]. Vision transformers have shown promise on large, diverse datasets, but often underperform CNN in small-to-moderate data regimes typical of clinical CT cohorts [16–18].

Reliable LVT segmentation in CT must be performed on contrast-enhanced images to effectively see the intrahepatic structures, such as vasculature and tumors. Training DL models on contrast-enhanced CT images on limited datasets is challenging due to the high image variability across contrast phases and patients. Recently, Random windowing was proposed as a CT-specific augmentation scheme

to expose the model to realistic phase variability due to contrast-enhancement [10, 11]. It samples clinically plausible HU windows stochastically during training, and has been shown to improve robustness in CT segmentation of liver tumors.

We aim to build on these aforementioned advances to make a unified LVT segmentation model from the available public datasets and demonstrate its potential for the clinic.

### 2.1 Learning from partial labels

As public datasets rarely share a complete LVT label space, combining them during training must be done with care to avoid class conflicts. A typical challenge is handling missing classes, because treating them as background in a softmax multi-class setup can cause conflicting regions. Prior work has addressed missing annotations via masked or partial-label losses that ignore unlabeled classes during training, preventing spurious gradients from missing annotations [19–21]. To this end, binary formulations with sigmoid outputs are particularly suitable, as they avoid the competition inherent in softmax and allow per-class supervision wherever labels are present [20, 22]. To mitigate noisy gradients from a multi-class segmentation setting using softmax, the loss masking must happen on a per-voxel basis, ignoring signals from all non-foreground regions [19].

Alternative strategies include weakly or semi-supervised learning using pseudo-labels [20, 23], and multi-task/multi-head designs [24, 25] where each dataset supervises a subset of heads while sharing an encoder. While effective in some settings, pseudo-labels can introduce confirmation bias from erroneous predictions, and multi-task heads can be difficult to calibrate across datasets.

Compared to approaches that stitch together separate binary models or rely on pseudo-labels to fill missing classes, we investigate segmentation models with a single segmentation head and an end-to-end training pipeline. In this setting, we investigate

how to benefit from and train on partially labeled datasets using various loss masking strategies.

## 2.2 Overlapping classes

Within the context of partial label supervision, overlapping classes are often treated as separate classes, masking the loss contribution of any missing class during training [21, 26]. Although possible in certain settings, it does not address the potential label conflicts across datasets, and could lead to suboptimal performance (Section 4.2).

Another established approach is cascaded models, which segment organ regions of interest (ROI) like the liver, before specialized models are trained for vessels and tumors within the organ ROI [9]. However, this approach leads to extra compute overhead during training and inference compared to an end-to-end pipeline in a complete label space.

Semantic segmentation on overlapping and partial labels has been addressed by probabilistic approaches that aggregate predictions [27] and multilabel approaches [28], but the approach within medical and LVT applications is largely underexplored.

## 2.3 Clinical potential for liver, vessel, and tumor segmentation

The clinical implications of LVT models are substantial. Below, we identify four routine scenarios where the LVT application has clinical potential: surgical planning, automatic liver volumetry, longitudinal patient follow-up, and tumor detection. In our final case study, we evaluate the current feasibility of LVT segmentation for each use case.

**Surgical planning** using accurate 3D spatial delineation of tumors to surrounding hepatic vasculature allows for more precise surgical planning. In an ongoing tangential study of the clinical impact of LVT models at the University Hospital of North Norway (UNN), initial results suggest significant improvements in surgical planning when a 3D LVT model was used along with CT and MRI images.

The integration of automatically generated 3D models into surgical procedures may significantly impact the management of complex hepatic resections [5]. 3D visualization of tumor-vessel relationships can aid surgeons when navigating in challenging anatomical landscapes [4, 7], and may reduce unintended vessel injury and improve resection margins.

**Automatic liver volumetry** further aids surgical decision-making by providing essential data for assessing hepatic functional reserve, which is crucial before for major liver resections [2, 3].

**During patient follow-up**, the LVT model has potential to benefit the clinical follow-up of oncology patients undergoing chemotherapy. By automatically measuring tumor sizes and the liver volume, the model can provide a consistent and objective assessment of tumor response over time [6]. This may facilitate timely therapeutic decisions, allowing clinicians to optimize the treatment based on tumor volume changes.

**Tumor detection** using a segmentation model enables precise identification of potential tumor regions in the CT scan. For patients at risk of liver metastasis, DL based tumor segmentation tools may help radiologists detect tumors early, improving diagnosis and treatment for patients.

**In general**, a performant LVT segmentation model has the potential to be a clinical tool and may improve diagnostic accuracy, enhance therapeutic planning, optimize patient follow-up, and increase surgical safety in hepatic tumor management.

# 3 Methodology

In this section, we present the baseline segmentation formulations and how we modify the setup and training loss to accommodate learning from partial labels. We present the masked loss formulations, before presenting how we handle label overlaps when learning from overlapping structures in the liver.

## 3.1 Training on partially labeled data

To enable training with partially labeled datasets, we investigate loss-masking of inconsistent signals in multi-class and multi-label segmentation.

*Multi-class* (MC) segmentation assumes mutually exclusive classes with one-hot encoded labels, and typically employs softmax activation function in the final layer. *Multi-label* (ML) segmentation treats each class independently using sigmoid activation, which allows overlapping regions through multi-hot encoded labels. If we leverage the ML architecture with the exclusive one-hot labels from MC, we refer to the setup as MLx.

In the partial label setting, computing the loss over regions with missing labels requires special care, as unlabeled ground truth could penalize valid predictions. To solve this, we formulate the objective functions as masked losses to zero out gradients from missing classes or ambiguous regions.

**Multi-label masked loss.** In the ML setting, the independence of class predictions allows us to selectively disable supervision for missing classes, without affecting others. We define a class-specific weight with values set to 0 or 1 depending on the available annotations for each dataset. Missing or unlabeled classes weigh the loss contribution to 0, while fully annotated classes with 1.

During training, for each class $k$ of a given sample with $N$ voxels, we compute the mean voxel loss $\ell_k$ and mask it with the per-class weight $w \in \{0, 1\}^K$.

The masked binary loss $\mathcal{L}_B$ for partially labeled samples is thus computed for each voxel $i$ as

$$\mathcal{L}_B = \frac{1}{\sum_{k=1}^{K} w_k} \left( \sum_{k=1}^{K} w_k \frac{1}{N} \sum_{i=1}^{N} \ell_{ik} \right). \qquad (1)$$

The normalization term $\sum w_k$ ensures the gradients are scaled dynamically based on the number of supervised classes, preventing magnitude shifts when switching between fully and partially labeled datasets.

**Multi-class masked loss.** In the MC setting, supervision is enforced across each voxel over all classes, using one-hot encoded labels and softmax outputs. The background is modeled explicitly in a separate channel, and any missing foreground label would be treated as background. To enable MC loss masking, the weight mask must therefore be enforced spatially to eliminate gradients from non-foreground regions not present in the partial labeled training set.

The per-voxel weight mask $W \in \{0,1\}^N$ enables the categorical loss $\mathcal{L}_C$ for a partially labeled sample

$$\mathcal{L}_C = \frac{1}{\sum_{i=1}^{N} W_i} \sum_{i=1}^{N} \ell_i W_i. \qquad (2)$$

Unlike the binary formulation, which normalizes by the count of active classes, $\mathcal{L}_C$ normalizes by the count of active voxels, essentially yielding the mean foreground loss. This eliminates the problem of ambiguous regions, also when the background is modeled explicitly.

## 3.2 Segmenting overlapping structures

Given complete annotations, class exclusive segmentation setup has the benefit of yielding unambiguous regions and explicit information about boundaries between classes. However, for partially labeled datasets, class exclusivity is not guaranteed, and overlapping classes risk regions of conflicting supervision due to overlap.

To avoid this issue across partially labeled datasets, ML can be trained with overlapping classes if labels are represented as multi-hot vectors, with per-voxel labels $y_{ik} \in \{0,1\}$ for each class $k$. This avoids competition between classes during training and allows supervision of whichever labels are available for that sample. In the context of partial labels, $w$ can seamlessly be applied in Equation 1 to mask the loss contribution of unlabeled classes.

**Anatomical liver hierarchy.** In the context of LVT segmentation, vessels and tumors are anatomically contained within the liver. In our following experimental settings, we enforce this anatomical hierarchy by mapping vessel and tumor label positives

into the liver channel: $y_{i,\mathrm{L}} \leftarrow y_{i,\mathrm{L}} \vee y_{i,\mathrm{V}} \vee y_{i,\mathrm{T}}$. Crucially, this ensures that the model learns to jointly predict all intrahepatic contents, and does not incur penalty if e.g. vessel labels are missing. This mapping is applied on-the-fly for datasets that provide one-hot labels, yielding a consistent multi-hot label space across datasets. When a class is not annotated for a sample, its loss weight $w_k = 0$, so it does not contribute to the objective.

# 4 Experiments

In this section, we study the presented categorical and binary loss formulations of Equation 2 and Equation 1 when training on partially labeled data using the MC and MLx setup. We also investigate how ambiguous regions from overlapping classes affect performance and how the anatomical hierarchy addresses this in the ML setup. We present the experiments and their results sequentially, and use the novel insight to inform our clinical case study.

**Experimental setup.** All experiments are performed under an identical medical image segmentation setup, where the objective is a segmentation map with mutually exclusive classes. We focus on end-to-end training pipelines using a U-Net-like architecture [29], building on modifications from nnU-Net [12] like deep supervision [30], and LeakyReLU activations [31] for more robust training and results. The model processes 3D patches of $128 \times 128 \times 96$ voxels, sampled from training images of $1 \times 1 \times 1$ mm voxel spacing. We leverage Random windowing [11] for joint CT preprocessing and intensity augmentation. Further details on the training and evaluation settings can be found in the Appendix A.2.

## 4.1 Learning from partial labels

We construct a controlled experiment simulating a training setting with multiple datasets with partial and missing labels. Specifically, we create partially labeled training sets from different partitions of one fully annotated source dataset. This lets us test and evaluate various approaches without considering noise from distribution shifts from other data and label sources.

**Simulating partial labeled training.** Based on the 303 images from the HepaticVessel dataset [8], with vessel and tumor segmentation labels, and the auxiliary liver segmentation labels from Tian et al. [32], we randomly sample 5 datasets of similar size. Specifically, 20 % fully annotated (LVT masks) are reserved as hold-out test set (HV test), 20 % are used as full supervision (FS) training with complete LVT annotations, 20 % have partial supervision (PS) with tumor mask only, 20 % with PS vessel mask only, and 20 % with PS from liver mask only. Note

that the liver mask comprises the complete liver organ, without "cutouts" for the vessel and tumor classes. In this regard, the liver overlaps with the vessel and tumor masks, similar to a real setting with partially labeled datasets. Further dataset details can be found in Table A.1.

The question we want to answer is "How can auxiliary datasets with partial labels enhance segmentation performance over only using the fully labeled training set?". To this end, we compare the MC and MLx end-to-end setups with their respective loss masking strategies with 25 % FS with and without auxiliary PS datasets. For reference, we also provide the 100 % FS training setting.

We report the mean segmentation performance on the HV test,the external Ircad [13], and Colorectal Liver Metastasis (CRLM) [33] test sets after 5-fold cross-validation training on the combined full and partially labeled data splits. We measure the Dice similarity coefficient (DSC) on the liver, vessels, and tumors of the respective test sets. Our full evaluation strategy can be found in Appendix A.2.3.

**Binary segmentation benefits from partial supervision.** Based on the results, presented in Table 1, we make the following observations:

(1) In the fully supervised settings, the segmentation DSC are comparable for the liver class of MLx and MC, and higher or on par for MLx on the vessel and tumor classes across both datasets.

(2) With partial supervision and multi-class segmentation, the DSC performance collapse compared to full supervision for almost all classes. However, the exception is segmentation performance on Ircad vessels, which exceeds all other settings. Upon closer inspection, the liver and vessel recalls of MC∩PS are actually the highest across all datasets, while the precision is lowest. This can explain the extremes in DSC, because it suggests that the model over-segments with many false positives. In the HepaticVessel dataset, it is a clear disadvantage as the vessel labels are minimal and to some degree lacking, but an advantage in the Ircad dataset, which has more dilated and detailed vessel structures. We suspect the cause of over-segmentation is the individual and unbalanced supervision each class receives in the masked loss of the categorical loss formulation. While an increase and drop in recall and precision, respectively, are observed also in the MLx∩PS setup, the DSC does not suffer as severely. We attribute this to the binary loss formulations, which natively balance foreground/background better, also in the partially labeled settings.

(3) Contrary to the MC setup, MLx benefits from the auxiliary partially labeled data in all settings. The results suggest that the masked binary loss formulation in the multi-label setup can learn from the available data, without interfering destructively with the unlabeled classes.

## 4.2 Training on overlapping classes

Although our desired output space is exclusive, with each voxel in the liver belonging to either the liver, vessel, or tumor class, it might be suboptimal and unnecessary during training. As binary outputs in the segmentation head allow multi-label training with overlapping classes, we investigate how the potentially conflicting regions across partially labeled datasets contribute to downstream performance.

In the same controlled environment as our partial label experiment, we ablate the effect of training on ambiguous regions. Specifically, we compare MLx and ML trained with one-hot and multi-hot labels (Section 3.2), respectively, in the FS and PS settings.

**Liver tumor segmentation is sensitive to label conflicts.** We report the segmentation DSC and the tumor detection sensitivity computed on the connected components of the predictions. Based on the results presented in Table 2, we make the following observations:

(1) Vessel segmentation is largely unaffected by the conflicting labels in the exclusive training setup. We suspect it to be a consequence of segmenting the small vessel structure in the comparatively large surrounding liver. As the vessel structures are small, the MLx model can learn to produce multi-label, rather than exclusive, class outputs without being punished significantly in the loss, as only the FS training set has complete labels with vessel "cutouts" that punish such behaviour.

(2) For the tumor DSC in the non-overlapping baseline, the performance is significantly worse compared to the overlapping version. Contrary to the vessel class, the tumor class is more massive, which leads to a larger loss impact when the model predicts the liver without cutouts for the FS set.

(3) The tumor detection sensitivity drops significantly as a consequence of partial supervision on ambiguous liver and tumor labels for HV test. The impact of this result is key, as it is not a matter of slightly worse or better segmentation overlap, but more liver tumors that are being detected. Lesion detection precision remains similar for both methods on HepaticVessel and CRLM test sets, and elevated for ML on Ircad.

## 4.3 Learning from public datasets with partial labels

For the clinical case study, we aim to build on public CT liver datasets with partial labels to scale up the training data. Leveraging the insights on PS with binary segmentation and anatomical hierarchy from our previous experiments, we train the LVT model under the multi-label, class-masked regime with overlapping classes described in Section 3.2. For a solid quantitative baseline, we evaluate against the

**Table 1.** Segmentation DSC reported on the liver, vessel, and tumor classes of the HepaticVessel test set and the external Ircad and CRLM test sets. We report the segmentation performance along with the proportion of full supervision (FS) in the training set, whether partial labeled datasets (PS) were used as auxiliary training signal, and the segmentation head used, multi-class (MC) vs. multi-label exclusive (MLx). We measure statistical significance (*) at $p < 0.05$ using Wilcoxon signed-rank test.

| FS | PS | Head | HepaticVessel Liver | Vessel | Tumor | Ircad Liver | Vessel | Tumor | CRLM Liver | Vessel | Tumor |
|---|---|---|---|---|---|---|---|---|---|---|---|
| 25 % | × | MC | $0.977 \pm 0.001^*$ | $0.579 \pm 0.010$ | $0.517 \pm 0.024$ | $0.951 \pm 0.003^*$ | $0.372 \pm 0.033$ | $0.484 \pm 0.013$ | $0.937 \pm 0.000^*$ | $0.579 \pm 0.005$ | $0.564 \pm 0.020$ |
|  | × | MLx | $0.975 \pm 0.002$ | $0.600 \pm 0.004^*$ | $0.536 \pm 0.009$ | $0.950 \pm 0.002$ | $0.439 \pm 0.012^*$ | $0.501 \pm 0.003$ | $0.936 \pm 0.001$ | $0.610 \pm 0.007^*$ | $0.613 \pm 0.016^*$ |
| 25 % | ✓ | MC | $0.940 \pm 0.002$ | $0.428 \pm 0.006$ | $0.221 \pm 0.026$ | $0.913 \pm 0.002$ | $0.489 \pm 0.026$ | $0.341 \pm 0.045$ | $0.897 \pm 0.003$ | $0.445 \pm 0.017$ | $0.394 \pm 0.032$ |
|  | ✓ | MLx | $0.977 \pm 0.001^*$ | $0.629 \pm 0.003^*$ | $0.561 \pm 0.018^*$ | $0.946 \pm 0.002^*$ | $0.466 \pm 0.011$ | $0.525 \pm 0.037^*$ | $0.937 \pm 0.001^*$ | $0.650 \pm 0.004^*$ | $0.640 \pm 0.015^*$ |
| 100 % | × | MC | $0.980 \pm 0.000^*$ | $0.638 \pm 0.008$ | $0.615 \pm 0.023$ | $0.948 \pm 0.001^*$ | $0.397 \pm 0.008$ | $0.462 \pm 0.017$ | $0.939 \pm 0.000^*$ | $0.618 \pm 0.003$ | $0.578 \pm 0.015$ |
|  | × | MLx | $0.978 \pm 0.001$ | $0.698 \pm 0.004^*$ | $0.790 \pm 0.010^*$ | $0.946 \pm 0.001$ | $0.468 \pm 0.010^*$ | $0.573 \pm 0.025^*$ | $0.938 \pm 0.001$ | $0.651 \pm 0.003^*$ | $0.697 \pm 0.005^*$ |

**Table 2.** We ablate the effect of ambiguous regions, due to partial labels, during training. By allowing overlapping classes through multi-hot encoded labels, the binary segmentation head avoids mixed signals from the partially labeled liver dataset (lacking vessel and tumor). Allowing overlapping classes in the partial supervision setting leads to improved tumor segmentation.

| FS | PS | LVT overlap | HepaticVessel Vessel | Tumor | Sensitivity | Ircad Vessel | Tumor | Sensitivity | CRLM Vessel | Tumor | Sensitivity |
|---|---|---|---|---|---|---|---|---|---|---|---|
| 25 % | × | × | $0.600 \pm 0.004$ | $0.536 \pm 0.009$ | $0.734 \pm 0.034$ | $0.439 \pm 0.012$ | $0.501 \pm 0.003$ | $0.639 \pm 0.031$ | $0.610 \pm 0.007$ | $0.613 \pm 0.016$ | $0.712 \pm 0.034$ |
|  | × | ✓ | $0.601 \pm 0.009$ | $0.535 \pm 0.016$ | $0.740 \pm 0.033$ | $0.437 \pm 0.015$ | $0.472 \pm 0.031$ | $0.663 \pm 0.037$ | $0.609 \pm 0.008$ | $0.613 \pm 0.017$ | $0.714 \pm 0.034$ |
| 25 % | ✓ | × | $0.629 \pm 0.003$ | $0.561 \pm 0.018$ | $0.779 \pm 0.027$ | $0.466 \pm 0.011$ | $0.525 \pm 0.037$ | $0.706 \pm 0.017$ | $0.650 \pm 0.004$ | $0.640 \pm 0.015$ | $0.770 \pm 0.013$ |
|  | ✓ | ✓ | $0.629 \pm 0.005$ | $0.611 \pm 0.013^*$ | $0.818 \pm 0.037^*$ | $0.462 \pm 0.011$ | $0.536 \pm 0.035$ | $0.714 \pm 0.065$ | $0.649 \pm 0.006$ | $0.649 \pm 0.008^*$ | $0.767 \pm 0.007$ |

**Table 3.** Evaluation of our multi-label segmentation network and the full-res nnU-Net trained on MSD Liver (liver + tumor) and MSD HepaticVessel (vessel). The models are evaluated on contrast-enhanced CT images from the UNN external dataset.

| Task | Metric | nnU-Net | LVT (ours) |
|---|---|---|---|
| Tumor | DSC | $0.723 \pm 0.145$ | $\mathbf{0.778 \pm 0.106}$ |
|  | NSD | $0.706 \pm 0.214$ | $\mathbf{0.771 \pm 0.136}$ |
| Liver | DSC | $\mathbf{0.912 \pm 0.07}$ | $0.898 \pm 0.066$ |
|  | NSD | $0.951 \pm 0.058$ | $\mathbf{0.959 \pm 0.051}$ |
| Vessels | DSC | $0.545 \pm 0.051$ | $\mathbf{0.575 \pm 0.048}$ |
|  | NSD | $0.788 \pm 0.061$ | $\mathbf{0.808 \pm 0.053}$ |

datasets with a multi-label, masked loss and Random windowing improves robustness and clinical relevance. We attribute the main driver of gains to the scaled up training data enabled by the multi-label masked loss. We next present our qualitative retrospective analyses in Section 5.

# 5 Case study

Up until this point, we have validated our methods from a quantitative perspective. In this section, we shift our focus to the clinical practice and highlight the clinical usefulness of the LVT application through a retrospective feasibility analysis on longitudinal clinical data.

## 5.1 Tumor detection

Retrospectively analyzing the longitudinal CT scans of a patient and comparing the predictions with the radiology reports from the follow-up allows us to identify if the model could have assisted in the early detection of tumors. Such retrospective analysis helps identify when what might seem like a false positive tumor prediction by the model actually was a missed tumor by the radiologist.

For a given patient surgically treated for colorectal cancer at UNN, with a high risk of developing liver metastasis, we obtained predictions for the contrast-enhanced liver CT scans from the follow-up studies in the patient pathway from both our LVT model and the nnU-Net baseline. After the patient's initial treatment, they had no metastasized liver cancer for the following 1.5 years, but in March 2009, a 4 cm tumor was discovered in the left liver lobe. In the preceding CT scan, 6 months before, the radiologist

strong, but specialized nnU-Net.

We scale up partial labeled training using the complete HepaticVessel dataset (vessel + tumor) with additional liver labels from [32], the LiTS dataset [9] (liver + tumor), and Ircad (liver, vessel, tumor). We use Random windowing [11] for CT intensity augmentation and to mitigate cross-dataset shift. For all other training configurations, we follow the nnU-Net setup. Additional dataset and training details can be found in Appendix A.1 and A.2.

For quantitative evaluation, we compare against full-resolution nnU-Net baselines trained on MSD Liver (liver+tumor) and MSD HepaticVessel (vessel). We report DSC and normalized surface dice (NSD) on an external test set of contrast-enhanced CT images from UNN in Table 3.

Our model outperforms nnU-Net on tumors and vessels across DSC and NSD, and achieves higher liver NSD with slightly lower liver DSC, consistent with minor over-segmentation addressed by the surface-tolerant NSD metric. These results indicate that learning from additional partially labeled

Early detection of liver metastasis

September 2008          March 2009

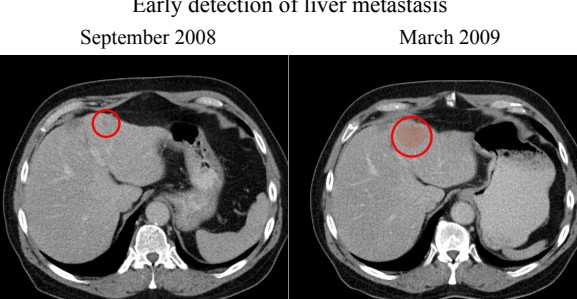

**Figure 2.** Comparison of CT images of the same patient 6 months apart. In the preceding scan from September 2008, the radiologist identified no suspicious lesions in the liver. 6 months later, the radiologist found a tumor measuring 4 cm in diameter. Our LVT model marked a corresponding lesion displayed in the image in the former image, 6 months before the radiologist.

stated that there were no suspicious lesions in the liver. However, retrospective analysis with our LVT model marked a small tumor region in the left liver lobe in the same scan, 6 months prior to the radiologist identifying a liver tumor in that same region (Figure 2). The prediction from the nnU-Net did not flag this region, likely due to limited diversity of its training set compared to the LVT model.

The flagged region was later confirmed by clinical liver expert and co-author MD K.R. as a plausible missed tumor due to its characteristics and location. However, confirming this with absolute certainty is difficult due to the retrospective nature.

Nonetheless, the example demonstrates the potential feasibility of DL assisted image analysis and how it could have led to significantly earlier detection of liver metastasis if it was used during follow-up of the patient.

## 5.2 Follow-up and tumor monitoring

A key consideration when treating a patient with, e.g., liver metastasis, is the size of the metastatic region over time. Tracking the lesion's size helps assess how the patient responds to the treatment they receive. Decreasing tumor size suggests that the patient responds well to the treatment, while growth indicates tumor resistance to the treatment.

We retrospectively analyze a patient's CT liver scans during the follow-up period and assess how the LVT model performs automatic size measurements of the tumor. We report the largest dimension of the tumor in the x-y plane and compare it to the radiologist's measurements at the time of the study.

We present the results in Figure 3 and find the extracted measurements to correlate well with the radiologist's measurements. During the follow-up period, the patient experienced an initial period of

Predicted vs reference tumor sizes over time

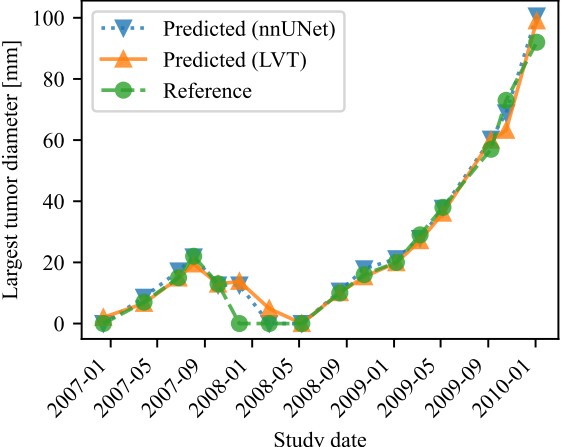

**Figure 3.** Comparison of model vs. radiologist measurements for a patient with metastasis, tracking initial treatment response followed by progression.

tumor growth after a lesion of liver metastasis was located in liver segment 4. The radiologist's and the model's predictions align well during this period. After the initial tumor growth, the patient was considered inoperable and began chemotherapy with a good response, leading to tumor regression. However, in two scans during the regression, the model yielded false positives in another liver segment, compared with the radiologists' findings. Six months post-treatment, disease progression was again observed, and despite further management, the malignancy continued to progress. During this critical period, the LVT model matches the radiologist in tumor detection and size prediction.

## 5.3 Surgery planning

Surgical resection of tumors is, in most cases, considered the only cure for liver metastasis [1]. Due to the complex hepatic vasculature the tumor's precise location and relation to surrounding vessels are crucial. Since manual segmentation of these structures is expensive and rare in clinical practice, automatic segmentation provides novel insight for the multidisciplinary team and surgeons treating the patient.

Our LVT segmentation model is able to precisely delineate the tumor and blood vessels in high-quality contrast-enhanced CT images of the liver. For a patient at UNN, we retrospectively obtain LVT predictions from their CT images to illustrate the output when visualized in 3D software. The results are shown in Figure 4 and display the liver and delineation of a liver tumor in segment 7 with its surrounding vessels. The 3D view of the LVT predictions makes the evaluation of proximity to the structures surrounding the tumor. The visualizations are produced with 3D Slicer image computing

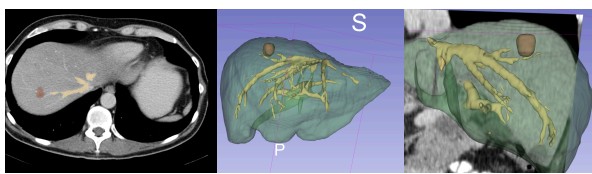

**Figure 4.** Illustration of automatic 3D segmentation of liver, vessels, and tumors in a CT image.

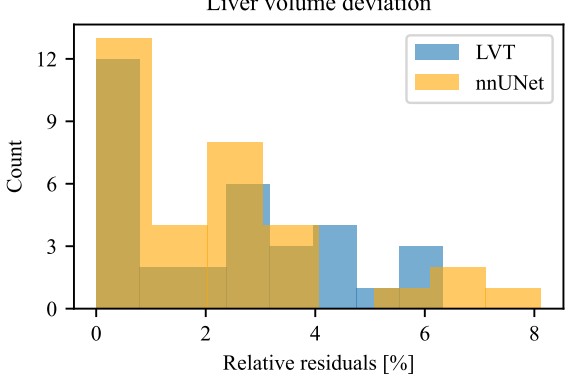

**Figure 5.** Relative deviation of estimated liver volume between images of different contrast phases.

platform [34], where the user can edit the predictions if needed, and subsequently make precise measurements before final assessment.

## 5.4 Automatic liver volumetry

Prior to major liver resections, CT volumetry can be used to measure the liver volume and estimate the hepatic functional reserve for the patient.

We aim to illustrate the efficacy of the LVT model for this purpose in a clinical setting by retrospectively analyzing unlabeled images from the clinic. We assume that a patient's liver on images from the same day is similar in size, and we aim to compare the model's predictions across contrast phases. To this end, we retrieved 33 images from 14 patient studies, where there are 2 or 3 contrast-enhanced images in each study, with images in the arterial, venous, or late phase. We use the LVT model to obtain liver masks for each patient and compute the liver volumes for each image. As the liver sizes vary from 1.3 L to 2.5 L across patients, we report the relative residuals in percentages of the mean liver volume of each study shown in Figure 5.

We find the liver volume deviation from the reference to be within $\approx 5\%$ for most cases, and show that the liver measurements are consistent across images of the same patient. 5 % deviation is within the margin of what is expected from intra-observer variability [3]. Additionally, the measured volume is expected to vary slightly between images of different contrast phases [3].

## 6 Limitations and future work

Despite recent efforts, consistent and reliable segmentation of liver tumors and vessels remains a difficult task. Table 3 reports improvements over the baseline, but DSC scores remain below 0.8 for tumors and vessels, which suggests that false positives and missed structures are to be expected. False positives may arise when cysts, necrotic tissue, or low attenuation liver regions are misclassified as tumors, or when blurred vessel boundaries lead to over-segmentation. Conversely, missed structures may occur for small and early-stage tumors, and poorly contrasted or thin vessels.

In the partial label experiments in Section 4.1 and 4.2, we observed certain inconsistencies in vessel segmentation performance across the HV Test set and Ircad dataset. These inconsistencies are symptoms of different label characteristics of the vessels in the two datasets, which have different quality and level of detail. We therefore recommend careful evaluation when comparing these datasets.

While our retrospective analyses demonstrate the potential for clinical utility of the LVT model, integrating it into real-time clinical workflows remains an open challenge. To further identify the strengths and limitations of DL-based liver, vessel, and tumor applications, we recommend thorough clinical validation with expert supervision to validate the model's impact on patient outcomes and clinicians' workflows.

## 7 Conclusion

This study explores multi-label and multi-class approaches in the context of CT liver, vessel, and tumor segmentation, to effectively handle overlapping and potentially ambiguous regions from partially labeled datasets. We find a binary multi-label segmentation setup with class-wise loss masking to work well for this setting. Allowing overlapping regions in the label space enables the use of public datasets with partial labels during training to learn simultaneous liver, vessel, and tumors labels in CT images. Our results show that our approach is particularly beneficial for tumors and vessels, allowing us to benefit from datasets with partial and ambiguous labels.

We evaluate the LVT model on longitudinal clinical data to illustrate the potential for real-world utility in the clinic. In retrospective analysis of previous patients, we demonstrated that the model has the potential to detect tumors earlier than the radiologist, accurately track tumor progression, provide 3D visualization of complex liver structures, and reliably perform liver volumetry for real patients. These results underscore the potential for AI-driven tools for diagnostic accuracy, optimizing treatment planning, and improving patient outcomes.

## Acknowledgments

Author EAØ thanks the Fine Grained Analysis group at the P1 Centre in Copenhagen for their warm hospitality and valuable discussions during his visit in 2024. The authors also thank Kim Erlend Mortensen and Eirik Kjus Aahlin for providing access to the UNN data.

This work was supported by The Research Council of Norway [Visual Intelligence, grant no. 309439 as well as FRIPRO grant no. 315029 and IKTPLUSS grant no. 303514]. The work was furthermore supported by Pioneer Centre for AI, DNRF grant number P1. We acknowledge Sigma2 – the National Infrastructure for High Performance Computing and Data Storage in Norway, for awarding this project access to the LUMI supercomputer, owned by the EuroHPC Joint Undertaking, hosted by CSC (Finland) and the LUMI consortium through Sigma2 [grant no. NN8106K].

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

# A  Appendix

## A.1  Datasets

In our experiments, we leverage multiple datasets that are publicly available, in addition to external test data from UNN. The following sections describe the datasets used in our experiments.

### A.1.1  HepatiVessel dataset

The HepaticVessel dataset is from the Medical Segmentation Decathlon challenge [8] and consists of 303 portal-venous phase CT scans from the US. The dataset has an out-of-plane voxel spacing ranging from 0.8 to 8.0 mm. The images contain the liver with segmented liver tumors and vessel structures.

### A.1.2  HepaticVessel Liver dataset

Building on the HepaticVessel dataset, a supplementary label set[1] of the liver (HV Liver) and Couinaud segments of the liver was released by Tian et al. [32]. The dataset contains all the same images as the HepaticVessel dataset, but the additional liver and Couinaud segmentation masks are created independently. We leverage the additional liver masks from HV Liver together with the HepaticVessel dataset in our experiments.

### A.1.3  3D-ircadb-01 dataset

The 3D-ircadb-01-dataset[2] [13] (Ircad), contains 20 CT scans from France that are labeled with various organs, including liver, hepatic vessels, and any liver tumors. The scans from the IRCAD dataset are a subset of the LiTS dataset; however, with only the liver and tumor masks are present, and no vessel masks [9].

---

[1]Available at: https://github.com/GLCUnet/dataset
[2]Available at: https://www.ircad.fr/research/data-sets/liver-segmentation-3d-ircadb-01/

### A.1.4  Liver tumor segmentation (LiTS) dataset

The LiTS dataset [9] contains 131 segmented CT volumes from different patients. The CT scans are from 7 different institutions in Canada, the Netherlands, Germany, France, and Israel. The CT images are contrast-enhanced and captured in the portal-venous phase and have an out-of-plane voxel spacing ranging from 0.7 to 5.0 mm. All images contain a rough segmentation mask of the liver in addition to a radiologist's segmentation of any liver tumors. The liver tumors are both primary and metastatic from colorectal, breast, and lung primary cancers.

The LiTS dataset has 20 volumes (volumes 28-47) overlapping with the 3D-ircadb-01 dataset [13]. This subset contains the same segmented liver and tumor masks as LiTS, in addition to vessel masks, which are used in the LVT application.

### A.1.5  Colorectal Liver Metastasis (CRLM) dataset

The CRLM dataset [33] contains 197 cases of portal-venous contrast-enhanced CT scans of patients with colorectal liver metastasis. The CT images are labeled with segmentation mask of the liver, hepatic vessels and tumor. We leverage the CRLM dataset as a fully labeled external test set.

### A.1.6  UNN Dataset

The UNN Dataset is under development at The University Hospital of North Norway (UNN) and UiT The Arctic University of Norway and is used for evaluation in our final experiments. The dataset is from a large database of CT images from the follow-up period of 376 patients that were treated for colorectal cancer from 2006 to 2011 at UNN. From this database, we have created two labeled subsets: UNN LT, which contains liver and tumor masks, and UNN V, which contains liver and vessel masks. The former is used for external validation and testing of the liver and tumor segmentation performance of the model and consists of 18 contrast-enhanced CT volumes with segmented liver and liver tumor masks. UNN V contains 10 contrast-enhanced CT volumes of the liver with segmented liver vessels and is used to evaluate the vessel segmentation performance in Section 4.3.

### A.1.7  Partial labels in datasets

As we train on datasets with partial labels for certain experiments, we present an overview of the present label classes for each dataset and the class weights $w$ needed for Equation 1 is presented in Table A.1.

**Table A.1.** An overview of the different datasets with partial labels used in this paper and the corresponding class weights $w_k$ for classes *liver, vessel* and *tumor* in Equation 1. Datasets with unspecified $w$ are only used for testing purposes.

| Dataset | Liver | Vessel | Tumor | Images | Class weights, $w$ |
|---|---|---|---|---|---|
| FS LVT | ✓ | ✓ | ✓ | 61 | $[1, 1, 1]$ |
| PS L | ✓ | ✗ | ✗ | 61 | $[1, 0, 0]$ |
| PS V | ✗ | ✓ | ✗ | 61 | $[0, 1, 0]$ |
| PS T | ✗ | ✗ | ✓ | 60 | $[0, 0, 1]$ |
| CRLM | ✓ | ✓ | ✓ | 197 | – |
| IRCAD | ✓ | ✓ | ✓ | 20 | – |
| HV test | ✓ | ✓ | ✓ | 60 | – |
| LiTS | ✓ | ✗ | ✓ | 131 | $[1, 0, 1]$ |
| HepaticVessel | ✗ | ✓ | ✓ | 303 | $[0, 1, 1]$ |
| IRCAD | ✓ | ✓ | ✓ | 20 | $[1, 1, 1]$ |
| HV Liver | ✓ | ✗ | ✗ | 303 | $[1, 0, 0]$ |
| UNN LT | ✓ | ✗ | ✓ | 18 | – |
| UNN V | ✗ | ✓ | ✗ | 10 | – |

## A.2 Experimental setup

All models in this paper are trained on 3D patches of $128 \times 128 \times 96$ voxels, sampled from training images resampled to isotropic voxel spacing of $1 \times 1 \times 1$ mm using trilinear interpolation. The U-Net-like architecture uses deep supervision [30] with two auxiliary heads at intermediate resolutions and LeakyReLU activations [31]. During training, patches are over-sampled from a foreground region with $p = 0.333$, and we apply the following augmentations in sequence: random crop resizing applied with probability $p = 0.2$ and a scale factor $\alpha \sim U(0.7, 1.4)$, random rotation with $p = 0.2$ and angle $\beta \sim U(-30, 30)$, and random flip with $p = 0.5$ along all axes. We leverage Random windowing [11] for preprocessing and CT intensity augmentation, applying window shifting and scaling independently with a total probability $p = 0.3$, sampling the Hounfield unit window parameters from $W \sim U(11.5, 152.9)$ and $L \sim [141.2, 325.9]$ [10]. Training is done with the combined CE and Dice loss (Equation 3 and Equation 4). All models are trained with a batch size of 112 images across 8 GPU compute dies on 4 AMD MI250x GPUs on the LUMI supercomputer.

The PS experiments in Section 4.1 and 4.2 are performed with a residual encoder [35] and batch normalization [36]. Training is done with AdamW [37] optimizer with learning rate 0.001 and cosine decay with warmup [37]. The models are trained 40 epochs with 100 steps each. The masking weights used for each dataset are listed in Table A.1.

The LVT model trained in Section 4.3 deviates from this setup to match the one used by the nnU-Net baseline. Specifically, we no residual connections in the encoder, instance normalization [38], stochastic gradient descent with weight decay optimizer [37] and polynomial learning rate decay. The model is trained on 448 000 training samples over 1000 epochs, which is comparable to the baseline, which

sees 500 000 training samples.

### A.2.1 Training loss

For each sample with $K$ classes, the cross-entropy loss is defined per voxel as

$$\ell_i^{CE} = -\sum_{k=1}^{K} y_{ik} \log p_{ik}, \qquad (3)$$

where $y_{ik}$ and $p_{ik}$ are the target and prediction from the one-hot encoded mask and softmax probabilities, respectively.

For the same network with sigmoid outputs, we obtain the case for binary cross-entropy, where the per-voxel per-class loss is defined as

$$\ell_{ik} = -[y_i \log p_i + (1 - y_i) \log(1 - p_i)]. \qquad (4)$$

The dice loss is computed independently for each voxel and class, given the output probabilities, and is given by

$$\ell_{ik}^{Dice} = 1 - \frac{2 \cdot y_{ik} p_{ik}}{y_{ik}^2 + p_{ik}^2}. \qquad (5)$$

In the multi-class segmentation setup, it is typically reduced over the class dimension to match the per-voxel loss formulation of the $\ell_i^{CE}$.

### A.2.2 Inference settings

As our models are trained on crops smaller than a typical CT image, we follow the sliding window inference pipeline of Isensee et al. [12] to obtain predictions. Specifically, each test volume is cropped into patches of $128 \times 128 \times 96$ voxels, with 50 % overlap. The model predictions on each patch are aggregated to a complete output with a gaussian weighing, as the predictions are usually more stable towards the center. The final semantic output is obtained through the argmax across channels. For the binary segmentation outputs, we use a sigmoid threshold of $p = 0.5$, and obtain mutually exclusive outputs by giving positives of the overlapping classes priority based on the heuristic hierarchy: tumor, vessel, liver, background.

For comparison with the nnU-Net in Section 5, we use the 5-fold cross-validation models to obtain an ensemble prediction of each pred. During inference, we use test-time augmentation by flipping each crop along all axes. We also limit the final prediction to the largest connected component. These inference settings are also employed by the baseline.

### A.2.3 Evaluation

**Precision and recall** are common metrics for evaluating classification performance using the true

positives ($TP$), false positives ($FP$), and false negative ($FN$) predictions. Precision measures the proportion of predicted positives that are correct:

$$\text{Precision} = \frac{TP}{TP + FP}, \tag{6}$$

while recall (sensitivity) measures the proportion of actual positives that are correctly identified:

$$\text{Recall} = \frac{TP}{TP + FN}. \tag{7}$$

Although pixel-wise precision and recall are not commonly used to evaluate segmentation masks, they can assist in diagnosing under and over-segmentation in models. Specifically, low recall tends to correspond to under-segmentation, and low precision to over-segmentationMonteiro and Campilho [39].

**Dice similarity coefficient (DSC).** To evaluate segmentation predictions against ground truth masks more reliably, we rely on the DSC, which measures volume overlap between predicted and true masks, $X$ and $Y$, as the harmonic mean of the precision and recall:

$$\text{DSC} = \frac{2|X \cap Y|}{|X| + |Y|}. \tag{8}$$

**Normalized surface dice (NSD).** While widely used, DSC treats all pixel errors equally, which may obscure clinically important mistakes (e.g., missing an entire tumor vs. scattered noise). Therefore, we additionally leverage NSD [40] in our clinical evaluation. NSD addresses this by comparing surfaces within a tolerance, defined per class in millimeters. Errors inside the tolerance do not reduce the score, making NSD more clinically meaningful. Following [8], we use 7 mm tolerance for liver and 3 mm for vessels and tumors.

**Lesion sensitivity.** Based on a connected component analysis of the ground-truth and predicted tumor segmentation, we classify a tumor in the ground truth as detected if they have a corresponding prediction with $> 10\,\%$ overlap. Based on this classification, we can compute the lesion recall/sensitivity using Equation 7.

**Reporting** In most evaluations, we report the mean result for each model in the 5-fold cross-validation evaluation. To showcase the variation between multiple runs of comparable methods, we use the standard deviation of performance between runs. The result in Table 3 deviates from this protocol, as the whole ensemble is used to obtain each prediction. We therefore report the per-case mean and standard deviation for this result.

**Statistical testing** When measuring statistical significance, we use Wilcoxon signed-rank test using pairwise metric performance across cases and folds. We indicate statistical significance (*) at $p < 0.05$ when a result is better than the comparable alternative.

## A.3 Case study details

Below we provide additional details on how patients were selected for the case study in section 5.

For Section 5.1, we manually screened the retrospective Hospital1 database (radiology reports and images) for a patient with a single liver lesion emerging during follow-up, with artifact-free portal-venous CTs at regular intervals and detailed reports. Additionally, we looked for cases where a preceding scan was flagged with a lesion by the LVT model, and the first patient meeting these criteria was used. The "early detection" judgment was based on image co-registration and qualitative review by a liver-expert and co-author MD K.R. The patient in Section 5.2 was selected for appropriate pathology, image quality, and consistently documented radiologist measurements. Section 5.3 shows the first patient from the UNN V dataset. In Section 5.4 we use all artifact free images with appropriate ROI and imaging protocol from the database.

