# OpenReview forum: "Liver, vessel, and tumor segmentation from partially labeled CT and multi-label masked learning"
_NLDL.org/2026/Conference — NLDL 2026 Poster_

### Official Review · Reviewer_GzSH · 2025-10-06
**Multi-label and multi-class segmentation of LVT from CT data**

**Rating:** 2
**Confidence:** 3

**Summary:**

The paper investigates multi-label and multi-class segmentation of Liver, Tumor and Vessel form partially-labeled CT datasets while handling the overlapping of objects by loss-masking per class/voxel to ignore missing annotation, and by using multi-hot encoding and inherent anatomical hierarchy. The experiments simulated partial labelling for training instead of using multi-class labels, which successfully improve the results. Additional case-study highlights the potential of clinical application of the solution and motivation behind it.

**Strengths:**

- clear motivation for clinical usage,
- proposed solution in certain cases significantly improved DSC
- interesting findings and observation described in fourth section,
- proposed solution achieved higher than nn-Unet baseline
- Clinical study highlights the benefit and motivation behind proposed solution

**Weaknesses:**

- in Eq1 it looks like we can remove part (sum(w_k)^()_()) appears in the same form in both divisor and denominator, it looks like there may be a mistake in eq or it should be written differently, shouldn't it reflect the number of classes that are available ? what if some are masked out and equal 0, they weights are equal 0 to normalize correctly ?
- there is incoherence in normalization term between eq 1 and 2 probably coming from mentioned above.
- plenty of important information is placed in appendix such as datasets description which should be in main part of the paper for its proper understanding,
- the deep learning-related novelty is limited, and majority of information required to reproduce the results are placed in appendix
-

**Justification:**

The article is interesting but looks more like a journal paper than the conference paper. The focus of the paper is on explaining certain parts behind medical part and its potential contribution, where the deep learning part is left behind or explained in Appendix, thus the novelty itself is hidden and not explored.

---

> ### Author Rebuttal · Authors · 2025-10-17
>
> Thank you for your constructive feedback. We hope the following rebuttal will help clarify and address your questions and concerns.
>
> **Notation in Equation 1**\
> Note that there is a nested sum in Equation 1, so that $\sum_{k=1}^K w_k$ will not cancel out in this case. However, we agree that at first glance it can look this way. To make it more explicit, we will therefore add parenthesis in the revised version to avoid this confusion:
> $$
> L_B = \frac{1}{\sum_{k=1}^K w_k} \left( \sum_{k=1}^K w_k \frac{1}{N} \sum_{i=1}^N l_{ik} \right)
> $$
>
> **Normalization in Equation 1 and 2**\
> Regarding the normalization in Equation 1 vs. 2, the difference is intentional, because the loss masking acts over different axes. To be clear, Equation 1 masks the loss per class, while Equation 2 masks over voxels. We will make this explicit to avoid the impression of inconsistency. In the case where $w_{k} = 0$ for all $k$ classes and $\sum_{k=1}^K w_k = 0$, we do not have a valid partially labeled training set as no foreground classes are present.
>
> **Novelty and use of appendix**\
> We would like to clarify that the main contribution of our work is not to provide large methodological contributions on the deep learning side, but instead focuses on the analysis of how we can effectively leverage the information available in the particular application domain of CT image segmentation (see line 90-100). Our analysis provides new insights and thus aligns well with NLDL's call for paper, which asks for "new and original research on all aspects of Deep Learning", “Applications”, and “Deep Learning for Sciences” (including medicine). We have therefore directed the reader to the appendix for some of the details on the methodology (learning rate, dataset details, etc.). However, we acknowledge that some of these details can also be moved to the main text. We will therefore in the camera-ready version move relevant details from Appendix A.1 and A.2 into section 3 and 4.

---

### Official Review · Reviewer_ecTN · 2025-10-07
**Solid execution, limited novelty, and overstated clinical claims**

**Rating:** 2
**Confidence:** 5

**Summary:**

The paper addresses an important challenge in medical image analysis: segmentation of liver, vessels, and tumors from partially labeled CT datasets. The authors propose a single-head, 3D multi-label segmentation network trained with a masked binary loss to ignore missing labels and multi-hot encoding to handle overlapping structures. They also integrate Random Windowing augmentation to mitigate contrast variability in CT scans.

While the study demonstrates that partial-label training can be achieved within this framework, the methodological novelty is limited. The paper’s framing as “towards clinical application” overstates the evidence, as the clinical analysis is anecdotal and lacks transparent case selection.

**Strengths:**

* Addresses a relevant problem, the scarcity and fragmentation of labeled LVT datasets are clearly articulated.
* Sound methodological setup. The use of masked binary loss and multi-hot label encoding is appropriate for overlapping classes and partially labeled data.
* Mostly thorough experimental structure. The authors present controlled ablations that separate fully and partially labeled regimes and evaluate their proposed approach (MLx) to a baseline (MC) .
* The introduction and related work sections are well organized and informative, placing the study in context of prior segmentation efforts.
* Figure 1 effectively illustrates the core idea.

**Weaknesses:**

* The title “Towards clinical application” is overstated and suggests translational progress beyond what is demonstrated. The clinical case studies are anecdotal and qualitative, not actual clinical validation.
* Sections 3.1–3.2: The masked-loss formulation and overlapping-label explanation are written with excessive formality, making a simple idea harder to follow than necessary.
* In the final evaluation, the proposed model combines masked multi-label training with Random Windowing augmentation, yet no ablation disentangles their individual effects. Even if the authors view this combination as their main contribution, they should clarify which element drives the observed gains.
* The reported performance differences are not supported by statistical testing or uncertainty analysis.
* Non-existent contribution in “Surgery planning.”, Section 5.3 merely shows that the model’s outputs can be visualized in 3D Slicer without any expert evaluation or quantitative assessment of utility.
* The qualitative examples in Sections 5.1–5.4 are insufficiently documented. It is not stated whether they come from the same patient, how they were chosen, or whether they are representative.
* While conceptually useful, Figure 1 could benefit from improved visual hierarchy and clearer data flow to enhance understanding.

**Justification:**

This work targets a clinically relevant problem (leveraging partially labeled data for LVT segmentation) and is somewhat competently executed. However, the methodological contribution is incremental, and the evidence provided is insufficient. The combination of masked multi-label learning with Random Windowing is not isolated experimentally (with appropriate ablations). No statistical testing or uncertainty estimation is provided. Finally, the clinical sections are anecdotal and lack methodological transparency. Thus, the paper falls short of novelty, experimental rigor, and substantiated impact required for acceptance.

---

> ### Author Rebuttal · Authors · 2025-10-22
>
> Thank you for your thorough review and for acknowledging the strengths of our contribution. In your constructive review, you bring up several aspects that we agree to revise for clarity.
>
> **Clinical positioning**\
> While we argue that demonstrating clinical potential, on real patient data, is an initial step towards a clinical application, we acknowledge that our initial title could seem misleading. Our main contribution lies with the study of partial label training, and we will thus reframe the clinical positioning of our paper. Highlighting the main contribution, we propose the new title: "Liver, vessel, and tumor segmentation from partially labeled CT and multi-label masked learning". Our clinical objective was to illustrate the potential of an integrated LVT model in a clinical setting, and we will explicitly reframe Section 5 as a "retrospective feasibility analysis on clinical data", to highlight this, and avoid overstating the evidence.
>
> **Documentation of case study**\
> The qualitative cases in Section 5.1-5.3 are from different patients whose cases illustrate the clinical feasibility of an integrated LVT model. The cases represent typical patient pathways, rather than being statistically representative samples. For transparency, we will add a paragraph in the appendix explaining how the patients were chosen.
>
> > For Section 5.1, we manually screened the retrospective Hospital1 database (radiology reports and images) for a patient with a single liver lesion emerging during follow-up, with artifact‑free portal‑venous CTs at regular intervals and detailed reports. Additionally, we looked for cases where a preceding scan was flagged with a lesion by the LVT model, and the first patient meeting these criteria was used. The “early detection” judgment was based on image co‑registration and qualitative review by a liver‑expert MD co‑author. The patient in Section 5.2 was selected for appropriate pathology, image quality, and consistently documented radiologist measurements. Section 5.3 shows the first patient from the ExDS V dataset. In Section 5.4 we use all artifact free images with appropriate ROI and imaging protocol from the database.
>
> **Methodology presentation**\
> In Section 3.1-3.2 we have presented the relevant strategies both conceptually and formally. Although our approach is intuitive to understand, we included the mathematical formulations as they are more precise than conceptual explanations. To prevent the formality to obstruct the main concept we moved the loss formulations to the appendix (Eq. 3-5) focusing only on partial label masking in Equation 1 and 2. For improved clarity, we will revise the first paragraphs of Section 3.1 and 3.2 to focus on the intuitive aspects of our method, before presenting the mathematical formulation improved with parenthesis for clarity.
> $$
> L_B = \frac{1}{\sum_{k=1}^K w_k} \left( \sum_{k=1}^K w_k \frac{1}{N} \sum_{i=1}^N l_{ik} \right)
> $$
>
> **Contribution of Random windowing**\
> As the main contribution of our paper is to address a clinically relevant application (liver, vessel, and tumor segmentation) and provide valuable and novel insights for training under partial label supervision, and handle overlapping anatomies, we do not ablate the effect of Random windowing. Random windowing is a conceptually simple and CT coherent way of doing augmentation of contrast-enhanced CT images, and is presented in a related paper. Random windowing has been shown to increase robustness in liver tumor segmentation and bridge cross-dataset distribution in CT images, and is used as is. Random windowing represents an incremental change towards a clinically geared application, as it is developed for the CT modality, rather than an adaptation of augmentation techniques for natural images. However, we acknowledge that it is a small piece of a larger pipeline, and we attribute the main driver of gains to the increased scale and training data from multiple sources. We will state this clearly in the revised and camera-ready version of our submission when presenting the role of Random windowing.
>
> **Statistical significance of results**\
> To address the reviewer's concern about statistical significance, we perform Wilcoxon's signed rank test between cases from all settings of our partial supervision experiments, further backing up our claims in the paper. Furthermore, we evaluate the models on an additional external dataset, the Colorectal Liver Metastasis dataset (CRLM) [1] (197 cases). The results on this external dataset further support our results presented in the paper. We will include the updated results in the camera-ready version of the paper, highlighting statistically significant results with $p < 0.05$ in the revised tables, and include the CRLM dataset description.
>
> **Table 1**
> |        |              |      |                   | HepaticVessel     |                   |                   | Ircad             |                   |                   | CRLM              |                   |
> |--------|--------------|-----:|:-----------------:|-------------------|-------------------|:-----------------:|-------------------|-------------------|:-----------------:|-------------------|-------------------|
> |   FS   |      PS      | Head |       Liver       |       Vessel      |       Tumor       |       Liver       |       Vessel      |       Tumor       |       Liver       |       Vessel      |       Tumor       |
> |  25 \% |   $\times$   |  MC  | 0.977 ± 0.001$^*$ | 0.579 ± 0.010     | 0.517 ± 0.024     | 0.951 ± 0.003$^*$ | 0.372 ± 0.033     | 0.484 ± 0.013     | 0.937 ± 0.000$^*$ | 0.579 ± 0.005     | 0.564 ± 0.020     |
> |        |   $\times$   |  MLx | 0.975 ± 0.002     | 0.600 ± 0.004$^*$ | 0.536 ± 0.009     | 0.950 ± 0.002     | 0.439 ± 0.012$^*$ | 0.501 ± 0.003     | 0.936 ± 0.001     | 0.610 ± 0.007$^*$ | 0.613 ± 0.016$^*$ |
> |  25 \% | $\checkmark$ |  MC  | 0.940 ± 0.002     | 0.428 ± 0.006     | 0.221 ± 0.026     | 0.913 ± 0.002     | 0.489 ± 0.026     | 0.341 ± 0.045     | 0.897 ± 0.003     | 0.445 ± 0.017     | 0.394 ± 0.032     |
> |        | $\checkmark$ |  MLx | 0.977 ± 0.001$^*$ | 0.629 ± 0.003$^*$ | 0.561 ± 0.018$^*$ | 0.946 ± 0.002$^*$ | 0.466 ± 0.011     | 0.525 ± 0.037$^*$ | 0.937 ± 0.001$^*$ | 0.650 ± 0.004$^*$ | 0.640 ± 0.015$^*$ |
> | 100 \% |   $\times$   |  MC  | 0.980 ± 0.000$^*$ | 0.638 ± 0.008     | 0.615 ± 0.023     | 0.948 ± 0.001$^*$ | 0.397 ± 0.008     | 0.462 ± 0.017     | 0.939 ± 0.000$^*$ | 0.618 ± 0.003     | 0.578 ± 0.015     |
> |        |   $\times$   |  MLx | 0.978 ± 0.001     | 0.698 ± 0.004$^*$ | 0.790 ± 0.010$^*$ | 0.946 ± 0.001     | 0.468 ± 0.010$^*$ | 0.573 ± 0.025$^*$ | 0.938 ± 0.001     | 0.651 ± 0.003$^*$ | 0.697 ± 0.005$^*$ |
>
> **Table 2**
> |       |              |              |               | HepaticVessel     |                   |               | Ircad         |               |               |        CRLM       |               |
> |-------|--------------|:------------:|:-------------:|-------------------|-------------------|:-------------:|---------------|---------------|:-------------:|:-----------------:|:-------------:|
> |   FS  |      PS      |  LVT overlap |     Vessel    |       Tumor       | Sensitivity       |     Vessel    |     Tumor     | Sensitivity   |     Vessel    |       Tumor       |  Sensitivity  |
> | 25 \% |   $\times$   |   $\times$   | 0.600 ± 0.004 | 0.536 ± 0.009     | 0.734 ± 0.034     | 0.439 ± 0.012 | 0.501 ± 0.003 | 0.639 ± 0.031 | 0.610 ± 0.007 | 0.613 ± 0.016     | 0.712 ± 0.034 |
> |       |   $\times$   | $\checkmark$ | 0.601 ± 0.009 | 0.535 ± 0.016     | 0.740 ± 0.033     | 0.437 ± 0.015 | 0.472 ± 0.031 | 0.663 ± 0.037 | 0.609 ± 0.008 | 0.613 ± 0.017     | 0.714 ± 0.034 |
> | 25 \% | $\checkmark$ |   $\times$   | 0.629 ± 0.003 | 0.561 ± 0.018     | 0.779 ± 0.027     | 0.466 ± 0.011 | 0.525 ± 0.037 | 0.706 ± 0.017 | 0.650 ± 0.004 | 0.640 ± 0.015     | 0.770 ± 0.013 |
> |       | $\checkmark$ | $\checkmark$ | 0.629 ± 0.005 | 0.611 ± 0.013$^*$ | 0.818 ± 0.037$^*$ | 0.462 ± 0.011 | 0.536 ± 0.035 | 0.714 ± 0.065 | 0.649 ± 0.006 | 0.649 ± 0.008$^*$ | 0.767 ± 0.007 |
>
>
> **Presentation of Figure 1**\
> To improve the clarity of Figure 1, we suggest to simplify the "output" of the model, removing the three arrows from "combined predictions" to make it clearer what the model outputs and better show what goes into the masked loss. To improve the visual hierarchy, we aim to reorganize the figure, from left to right, starting with datasets and ending with the masked loss on the right side.
>
>
> **References**: \
> [1] A. L. Simpson et al., “Preoperative CT and survival data for patients undergoing resection of colorectal liver metastases,” Sci Data, vol. 11, no. 1, p. 172, Feb. 2024, doi: 10.1038/s41597-024-02981-2.

---

### Official Review · Reviewer_Qn1g · 2025-10-09
**A setting for training on partially labelled datasets**

**Rating:** 4
**Confidence:** 4
**Final Rating:** 4
**Final Confidence:** 4

**Summary:**

The paper presents a way of liver, vessel and tumor segmentation training on partial datasets (a set of datasets where not all annotations contain all labels).  The focus lies on producing a model that can be clinically applied.  The authors propose a loss setting that enables learning both from partial labels and overlapping ones (eg vessels inside liver). In particular, they suggest using a multi-label binary segmentation with masked loss (ignoring voxel labels for absent classes), for training a simple (single-head/task) model in an end2end manner.
They first compare training a multi-class setting (with softmax) masking out unknown voxels (all classes), with training a multi-binary (multi-label) setting, masking out only the unknown classes each time. The experiments show the expected: sigmoids are better, full supervision is best.
Then they do the experiments whilst using overlapping classes (and on the fly hierarchy inclusion), comparing exclusive and nonexclusive multi-label training. Only tumors seem to be affected (improved) by the nonexclusive training.

**Strengths:**

The paper focuses on a problem that (especially in terms of clinical applicability) does not have a good enough solution yet. It uses a combination of newer and established ideas (random windowing augmentation, masking losses), and it evaluates the approach in terms of medically relevant outcomes, which is not so common in literature yet. One interesting point is that they address multiple things simultaneously - incomplete labellings (addressed by masking the loss), labels being hierarchical (if only vessel or tumor label exist, the liver label should be assigned at those pixels too, since those classes are part of liver), and inter- and intra-dataset differences in phase/contrast (addressed by using augmentation).
In experimental evaluation they compare both softmax och sigmoid-based (multi-label) training, with and without label overlap, and compare to baseline nnUNet. Evaluation is also done in terms of practical value for tumor detection and volumetry, though on a single patient.

**Weaknesses:**

While applying the approach to LVT segmentation may be unexplored in literature, the approach itself is not really new. Masked or partial label losses that ignore unlabelled classes during training is an established approach. And the windowing augmentation they use has been presented before.

The work is also not so clearly presented.  What do you mean by "fragmented", "LVC" for example - introduce that.  Considering eq 1 and 2, it should be made clear what each variable means (how exactly are the weights set - when is 0 weight used), and how the different experiments relate to eachother. (I was only clear on what all different experiments are being done after I read the full paper - consider adding a sentence or two before, explaining the three different experiments you run, make it clear which methods are used where and on which data when).

When comparing to nnUNet baseline (last table), it is not clear how that nnUNet is trained - how does it handle the missing labels etc. Is it trained simply with a 3-class softmax?  Or masking? Is augmentation applied there too, or only in your compared method? What is even your LVC method here - using LMx and overlapping or no overlapping? Be clearer on that, it is never properly defined.

I like the way you evaluate the approach in terms of practical medial applications, however it is not clear if any such evaluations have been used in literature before; is that your contribution too? When describing all the different medical applications where LVT could come in use (under sec 2.3) I would like to see actual past works on that; a mention on whether anyone has ever evaluated or considered such evaluations etc to get a feeling of whether it is reasonable.

For tumor detection, I am a bit weary of drawing conclusions from  the qualitative evaluation - that is a single patient case and it is not clear if you looked at any more patients (and how often it was both methods that didn't detect anything for example). Have you checked with a radiologist that the prediction actually is reasonable?  As for tumor growth, it looks like there may be some overestimation in size (especially during regression, when the size is getting smaller). This could mean even for your tumor detection results, that the segmented amount of early tumor is not realistic. So given that the accuracy of such unexpected detection cannot be established, I wonder how much practical value this has - if someone indicated that there perhaps is some tumor after all, could it be double checked or can it really not affect the workflow until the tumor presence and size can be safely established?

Figure 5: you say the liver volume residual deviations are within 2%, but it seems that quite a few are over that margin. Both for your method and nnUNet...?

**Final Justification:**

The paper does not present a novel method, but a useful collection of established modules.
What makes the pipeline useful (also medically) is that it addresses multiple things simultaneously - incomplete labellings (loss masking), labels being hierarchical (assigning parent labels), and inter- and intra-dataset differences in phase/contrast (augmentation). Dealing with hierarchical labels is an incremental and niche development, however it is of interest to see that their approach appears better than simply disregarding the nonexistent labels (which is done most commonly in literature).
While I agree that the evaluation (in terms of clinical value) is weak, since it is done on single, manually picked subjects only, and even then not clearly better than the alternative, the ways of evaluation are interesting for a broader public in clinically grounded AI.

**Justification:**

While the work does't present any specific novel method, but rather constructs a good combination of previous works for the specific problem, the evaluation is interesting and the combination of everything (taking into account hierarchy in missing overlapping labels, masking, and use of windowing) is clearly beneficial in the realistic case of combining datasets with missing labels. The more medically oriented evaluation is not thorough enough but it is a good idea probably of interest to others in applied medical imaging.

---

> ### Author Rebuttal · Authors · 2025-10-22
>
> Thank you for your thorough review.
>
> **Presentation**\
> Thank you for your comments on the presentation. We use the term "fragmented" when referring to the inconsistent and incomplete label space in datasets developed for CT liver segmentation applications. To avoid confusion and improve consistency, we will, in the revised manuscript, resort to the term "partially labeled data", and avoid the term "fragmented". We do not use the abbreviation "LVC" in our submission, but "LVT", when referring to "liver, vessel, and tumor" as defined at first occurrence (line 50).
>
> For Equations 1 and 2, all variables are defined in Section 3.1. As the loss formulations are more general than our specific application, we halt the presentation of how the weights are chosen until we explain the Anatomical liver hierarchy in Section 3.2. Further details on weight selection is presented in Table A.1. To avoid confusion, we intend to add a statement in Section 3.1 for clarity:
> > "The respective weights are set to 0 for ambiguous regions and missing classes".
>
> Thank you for the suggestion of outlining the experiments we perform. We will revise the intro of Section 4 to give the reader a heads up: "In this section, we study the presented categorical and binary loss formulations of Equations 1 and 2 when training on partially labeled data using the MC and MLx setup. We also investigate how ambiguous regions from overlapping anatomy affect performance and how the anatomical hierarchy can mitigate this in the ML setup."
>
> **Baseline and implementation details**\
> In Section 4.3, the experimental setup differs from the previous experiments, as we scale up training to 4 different partially labeled public datasets. For the nnU-Net baseline in Section 4.3, we use the official implementation and pretrained weights on the respective datasets. The official nnU-Net implementation does not support our exact partially labeled training setup, but represents a very strong baseline in the field of medical image segmentation. To make the comparison as fair as possible, we mimic the official nnU-Net training pipeline in architecture, hyperparameters, resources, inference, and disclose wherever we fail to meet their training recipe (due to our specific hardware limitations or multi-dataset training setup). To reiterate the details from the nnU-Net documentation and paper, the nnU-Net baselines defaults to softmax output – multi-class within each task, no loss masking, and intensity augmentations (contrast, brightness, gamma and inverse gamma) instead of Random windowing, and is trained on each dataset separately, not end to end.
>
> Thank you for pointing out the missing details of our implementation and baseline. The final LVT model use what we identify as the best configuration of the initial experiments, namely multi-label segmentation with overlapping classes following our described label hierarchy. We will specify this in the revised camera-ready version.
>
> **Clinical application and related work**\
> Thank you for the recognition of our work relating our study to clinical potential. We acknowledge that our case study could be further related to parallel research, positioning our case study better in the existing literature. In the revised and camera-ready submission, we will include additional context and work pulling in the same direction.
>
> The four use-cases we illustrate potential for are grounded in clinical tasks and prior literature. Certain clinical validations for automatic segmentation exist, with findings in agreement with our Section 5.2 and 5.4. However, to our knowledge, no prior LVT works evaluate all four axes in one system. We will make this explicit, better positioning our case study in existing literature.
>
> **Documentation of case study**\
> The qualitative cases in Section 5.1-5.3 are from different patients whose cases illustrate the clinical feasibility of an integrated LVT model. The cases represent typical patient pathways, rather than being statistically representative samples. For transparency, we will add a paragraph in the appendix explaining how the patients were chosen.
>
> > For Section 5.1, we manually screened the retrospective Hospital1 database (radiology reports and images) for a patient with a single liver lesion emerging during follow-up, with artifact‑free portal‑venous CTs at regular intervals and detailed reports. Additionally, we looked for cases where a preceding scan was flagged with a lesion by the LVT model, and the first patient meeting these criteria was used. The “early detection” judgment was based on image co‑registration and qualitative review by a liver‑expert MD co‑author. The patient in Section 5.2 was selected for appropriate pathology, image quality, and consistently documented radiologist measurements. Section 5.3 shows the first patient from the ExDS V dataset. In Section 5.4 we use all artifact free images with appropriate ROI and imaging protocol from the database.
>
> **Early detection and patient follow-up**\
> In the case of early tumor detection, the region flagged with early tumor detection by our model is confirmed as a plausible missed tumor by a clinical liver expert (MD and second-author). However, confirming this with absolute certainty is difficult due to the retrospective nature of our study. We will moderate our claims based on this finding in the camera-ready submission to avoid overselling this result.
>
> When it comes to patient follow-up and tumor growth, you are correct that both models overestimate the size of the tumor during regression, and we consider it a false positive. We have disclosed this in the results (Section 5.2) and as a limitation of the current work (Section 6). You raise a valid point about the practical value of such uncertain predictions in tumor detection. We do not claim to have the solution to this problem, but encourage further investigation in this direction to unlock the potential.
>
> **Figure 5**\
> Thank you for pointing out the inconsistency in the text vs Figure 5. The correct number, consistent with Figure 5, should be 5 \%. The revised statement for the manuscript (line 618-624) is:
> > We find the liver volume deviation from the reference to be within $\approx 5\%$ for most cases, and show that the liver measurements are consistent across images of the same patient. 5 \% deviation is within the margin of what is expected from intra-observer variability [1]. Additionally, the measured volume is expected to vary slightly between images of different contrast phases [1].
>
> **References**\
> [1] M. C. Lim, C. H. Tan, J. Cai, J. Zheng, and A. W. C. Kow, “CT volumetry of the liver: Where does it stand in clinical practice?,” Clinical Radiology, vol. 69, no. 9, pp. 887–895, Sept. 2014, doi: 10.1016/j.crad.2013.12.021.

---

### Official Review · Reviewer_arAZ · 2025-10-09
**CT segmentation using partially labeled data**

**Rating:** 2
**Confidence:** 4

**Summary:**

The study presents deep learning based approach using partial labelings for segmentation of structures from CT scans. Methods wise, the study does not seem to offer novelty, and the improvement in experimental results is incremental and apprears to be insignificant. The study also claims to present steps towards clinical use, which should more be framed as experimental case studies rather than clinically relevant evidence.

**Strengths:**

Study discusses a relevant problem of developing DL-based segmentation methods using partially labeled input data. Some of the experimental results show very high performance using numerical metrics.

**Weaknesses:**

Methodological novelty is limited, and improvement incremental and due to limited dataset, hard to confirm the significance. The article falls short in showing true clinical relevance despite raising this angle in the title.

**Justification:**

The article lacks convincing results towards its aim or significant novelty in methods.

---

> ### Author Rebuttal · Authors · 2025-10-17
>
> Thank you for your thoughtful review. We appreciate that you value the relevance of the problem we address.
>
> **Novelty** \
> We acknowledge the reviewer's comment on novelty. While we do not claim a new theoretical loss or architecture, the paper offers an original study with empirical insights aligned with NLDL's call on "new and original research on all aspects of Deep Learning" for papers on “Applications,” “Deep Learning for Sciences” (including medicine) and “semi-supervised learning”. We address a clinically relevant application (liver, vessel, and tumor segmentation) and provide valuable and novel insights for training under partial label supervision and handling overlapping anatomies.
>
> **Clinical Relevance** \
> Regarding the reviewer's concern in "showing true clinical relevance", we modestly frame the clinical angle as “towards" application, with retrospective evidence rather than deployment claims. The case study is retrospective and qualitative, but illustrate potential clinical utility, using original longitudinal clinical data not available to the broader machine learning community. We argue that demonstrating such potential, on real patient data, is an initial step towards a clinical application and goes considerably beyond the current practice of only providing some quantitative results on benchmark datasets.
>
> **Statistical significance of results** \
> To address the reviewers concern about statistical significance and limited datasets of our quantitative results, we perform Wilcoxon's signed rank test between cases from all settings of our partial supervision experiments, further backing up our claims in the paper. Furthermore, we evaluate the models on an additional external dataset, the Colorectal Liver Metastasis dataset (CRLM) [1] (197 cases). The results on this external dataset further support our results presented in the paper. We will include the updated results in the camera-ready version of the paper, highlighting statistically significant results with $p < 0.05$ in the revised tables, and include the CRLM dataset description.
>
> **Table 1**
> |        |              |      |                   | HepaticVessel     |                   |                   | Ircad             |                   |                   | CRLM              |                   |
> |--------|--------------|-----:|:-----------------:|-------------------|-------------------|:-----------------:|-------------------|-------------------|:-----------------:|-------------------|-------------------|
> |   FS   |      PS      | Head |       Liver       |       Vessel      |       Tumor       |       Liver       |       Vessel      |       Tumor       |       Liver       |       Vessel      |       Tumor       |
> |  25 \% |   $\times$   |  MC  | 0.977 ± 0.001$^*$ | 0.579 ± 0.010     | 0.517 ± 0.024     | 0.951 ± 0.003$^*$ | 0.372 ± 0.033     | 0.484 ± 0.013     | 0.937 ± 0.000$^*$ | 0.579 ± 0.005     | 0.564 ± 0.020     |
> |        |   $\times$   |  MLx | 0.975 ± 0.002     | 0.600 ± 0.004$^*$ | 0.536 ± 0.009     | 0.950 ± 0.002     | 0.439 ± 0.012$^*$ | 0.501 ± 0.003     | 0.936 ± 0.001     | 0.610 ± 0.007$^*$ | 0.613 ± 0.016$^*$ |
> |  25 \% | $\checkmark$ |  MC  | 0.940 ± 0.002     | 0.428 ± 0.006     | 0.221 ± 0.026     | 0.913 ± 0.002     | 0.489 ± 0.026     | 0.341 ± 0.045     | 0.897 ± 0.003     | 0.445 ± 0.017     | 0.394 ± 0.032     |
> |        | $\checkmark$ |  MLx | 0.977 ± 0.001$^*$ | 0.629 ± 0.003$^*$ | 0.561 ± 0.018$^*$ | 0.946 ± 0.002$^*$ | 0.466 ± 0.011     | 0.525 ± 0.037$^*$ | 0.937 ± 0.001$^*$ | 0.650 ± 0.004$^*$ | 0.640 ± 0.015$^*$ |
> | 100 \% |   $\times$   |  MC  | 0.980 ± 0.000$^*$ | 0.638 ± 0.008     | 0.615 ± 0.023     | 0.948 ± 0.001$^*$ | 0.397 ± 0.008     | 0.462 ± 0.017     | 0.939 ± 0.000$^*$ | 0.618 ± 0.003     | 0.578 ± 0.015     |
> |        |   $\times$   |  MLx | 0.978 ± 0.001     | 0.698 ± 0.004$^*$ | 0.790 ± 0.010$^*$ | 0.946 ± 0.001     | 0.468 ± 0.010$^*$ | 0.573 ± 0.025$^*$ | 0.938 ± 0.001     | 0.651 ± 0.003$^*$ | 0.697 ± 0.005$^*$ |
>
> **Table 2**
> |       |              |              |               | HepaticVessel     |                   |               | Ircad         |               |               |        CRLM       |               |
> |-------|--------------|:------------:|:-------------:|-------------------|-------------------|:-------------:|---------------|---------------|:-------------:|:-----------------:|:-------------:|
> |   FS  |      PS      |  LVT overlap |     Vessel    |       Tumor       | Sensitivity       |     Vessel    |     Tumor     | Sensitivity   |     Vessel    |       Tumor       |  Sensitivity  |
> | 25 \% |   $\times$   |   $\times$   | 0.600 ± 0.004 | 0.536 ± 0.009     | 0.734 ± 0.034     | 0.439 ± 0.012 | 0.501 ± 0.003 | 0.639 ± 0.031 | 0.610 ± 0.007 | 0.613 ± 0.016     | 0.712 ± 0.034 |
> |       |   $\times$   | $\checkmark$ | 0.601 ± 0.009 | 0.535 ± 0.016     | 0.740 ± 0.033     | 0.437 ± 0.015 | 0.472 ± 0.031 | 0.663 ± 0.037 | 0.609 ± 0.008 | 0.613 ± 0.017     | 0.714 ± 0.034 |
> | 25 \% | $\checkmark$ |   $\times$   | 0.629 ± 0.003 | 0.561 ± 0.018     | 0.779 ± 0.027     | 0.466 ± 0.011 | 0.525 ± 0.037 | 0.706 ± 0.017 | 0.650 ± 0.004 | 0.640 ± 0.015     | 0.770 ± 0.013 |
> |       | $\checkmark$ | $\checkmark$ | 0.629 ± 0.005 | 0.611 ± 0.013$^*$ | 0.818 ± 0.037$^*$ | 0.462 ± 0.011 | 0.536 ± 0.035 | 0.714 ± 0.065 | 0.649 ± 0.006 | 0.649 ± 0.008$^*$ | 0.767 ± 0.007 |
>
> **References**: \
> [1] A. L. Simpson et al., “Preoperative CT and survival data for patients undergoing resection of colorectal liver metastases,” Sci Data, vol. 11, no. 1, p. 172, Feb. 2024, doi: 10.1038/s41597-024-02981-2.

---

### Meta-Review · Area_Chair_d9DG · 2025-11-02

**Recommendation:** Accept (Poster)
**Confidence:** 5

**Metareview:**

## Summary

This work addresses the task of segmenting liver, vessels, and tumours from partially labelled CT datasets. The authors propose the use of a single-head 3D segmentation network that can be trained with masked binary loss that can exploit partial labels. The paper is motivated for clinical use, and experimental validation is convincing.

## Reviewer comments and rebuttal

The initial reviews of the work were mixed. While all reviewers recognized the usefulness of focus on using partial labels in medical image segmentation tasks, they also point out the paper over-claims its contribution, mainly from the clinical practice purpose. Other reviews also point out missing statistical tests, clarification on the mathematical formulations, and lack of methodological novelty. The last point on novelty though can be ignored in this case as the empirical validation is sufficiently well done as also recognized by one of the reviewers.

In their response, the authors have addressed all the major concerns raised by the reviewers. For example, by conceding to change the title and reframing Sec 5 as "retrospective feasibility analysis on clinical data". They have also included appropriate statistical tests in their response.

I encourage the authors to consider restructuring the manuscript so that important details are in the main paper and not in the Appendix as pointed out by one of the reviewers.

Overall, while the work does not propose any novel methods, that in itself is not a limitation. The problem they are addressing is a challenging one, and can be of interest to practitioners in medical image analysis communities. There are no factual errors, and the authors have addressed all the key, valid, concerns raised by the reviewers.

---

### Decision · Program_Chairs · 2025-11-05

**Decision:**

Accept (Poster)

**Comment:**

We recommend a poster presentation given the AC and reviewers recommendations.